# Role of MCC/Eisosome in Fungal Lipid Homeostasis

**DOI:** 10.3390/biom9080305

**Published:** 2019-07-25

**Authors:** Jakub Zahumensky, Jan Malinsky

**Affiliations:** Department of Microscopy, Institute of Experimental Medicine, Academy of Sciences of the Czech Republic, 14220 Prague, Czech Republic

**Keywords:** MCC, eisosome, microdomain, lipids, ergosterol, sphingolipids, phosphoinositides, regulation

## Abstract

One of the best characterized fungal membrane microdomains is the MCC/eisosome. The MCC (membrane compartment of Can1) is an evolutionarily conserved ergosterol-rich plasma membrane domain. It is stabilized on its cytosolic face by the eisosome, a hemitubular protein complex composed of *Bin/Amphiphysin/Rvs* (BAR) domain-containing Pil1 and Lsp1. These two proteins bind directly to phosphatidylinositol 4,5-bisphosphate and promote the typical furrow-like shape of the microdomain, with highly curved edges and bottom. While some proteins display stable localization in the MCC/eisosome, others enter or leave it under particular conditions, such as misbalance in membrane lipid composition, changes in membrane tension, or availability of specific nutrients. These findings reveal that the MCC/eisosome, a plasma membrane microdomain with distinct morphology and lipid composition, acts as a multifaceted regulator of various cellular processes including metabolic pathways, cellular morphogenesis, signalling cascades, and mRNA decay. In this minireview, we focus on the MCC/eisosome’s proposed role in the regulation of lipid metabolism. While the molecular mechanisms of the MCC/eisosome function are not completely understood, the idea of intracellular processes being regulated at the plasma membrane, the foremost barrier exposed to environmental challenges, is truly exciting.

## 1. Introduction

Biological membranes are segmented into coexisting lateral microdomains with specific structures and functions. This separation enables spatiotemporal segregation and coordination of various processes in a single continuous membrane. One of the best characterized membrane microdomains is the yeast plasma membrane compartment of the arginine permease Can1 (MCC; [1]), which accumulates a dozen integral membrane proteins. These include the nutrient transporters Can1, Fur4, Lyp1, Mup1, and Tat4 [1,2,3,4,5], and tetraspan proteins belonging to the Sur7 and Nce102 families [2,6,7]. The MCC adopts a characteristic furrow shape, which is stabilized by a large cytosolic protein scaffold, the eisosome [8,9]. The core constituents of the eisosome are homologous BAR (*Bin/Amphiphysin/Rvs*) domain-containing proteins Pil1 and Lsp1 [10,11], which associate with the plasma membrane at the bottom of the MCC furrows [9]. These proteins have been shown to induce membrane curvature both in vitro [11,12,13,14] and in vivo [13].

The existence of furrow-like structures analogous to the MCC/eisosome is widely conserved throughout fungi, microalgae and lichens [9,15]. However, their significance in different species varies. While disruption of the furrow-forming MCC/eisosome does not lead to a severe phenotype in *Saccharomyces cerevisiae*, the loss of Pil1 results in abnormal sporulation of *Schizosaccharomyces pombe* [16], and *lsp1*Δ*pil1*Δ mutants of the fungal pathogen *Candida albicans* exhibit defects in cell wall synthesis, septin and actin distribution, and cellular morphogenesis [17]. Furthermore, these cells have a decreased ability to form hyphae and invade the host organism effectively due to reduced resistance to oxidative stress and copper [18,19].

The lipid biosynthetic machinery is localized mostly at the endoplasmic reticulum (ER). Most of the sensory apparatus regulating the homeostasis of these lipids (via activation of transcriptional and post-transcriptional mechanisms) is also localized at the ER membrane [20]. However, as the plasma membrane represents the vanguard of cellular defence against stress conditions, it is of great advantage to localize a lipid homeostasis sensory organ there as well. Increasing experimental evidence shows that the plasma membrane microdomain MCC/eisosome is an ideal candidate for such a sensor. Besides having distinctive lipid composition and a typical shape of a 200–300 nm long and 50 nm deep furrow [3,9,14,21], the microdomain accumulates various proteins in a dynamic fashion in response to nutrient availability, cell metabolism, membrane lipid composition and transmembrane potential [1,2,3,6,7,22,23]. Furthermore, the microdomains are oriented randomly and distributed quite uniformly throughout the plasma membrane, which allows them to sense local membrane disturbances and deformations. Specifically, mechanical stretching of the plasma membrane leads to flattening of the MCC/eisosome and the concomitant release of signalling molecules [22]. In addition, numerous studies documented their involvement in membrane adaptation also to other kinds of stress [18,19,24,25]. All these properties predispose the MCC/eisosome to transduce diverse stimuli to the cell interior and take part, via conserved signalling pathways, in the regulation of lipid homeostasis, which plays a prominent role in the stress response in general, and membrane adaptation in particular.

This review aims to systematize and synthesise the so far fractionated evidence about this aspect of MCC/eisosome function. The distinctive lipid composition of the MCC microdomain, as well as condition-specific presence of regulatory proteins within the eisosome, indicate the involvement of the MCC/eisosome in the homeostasis of different lipid species. The following sections are organized accordingly. Since the vast majority of knowledge regarding the involvement of the MCC/eisosome in lipid homeostasis has been gained thanks to studies of *S. cerevisiae*, we have used the nomenclature of this species in the following study. Nevertheless, the pathways and mechanisms described there are, to a high extent, conserved in other fungi, or at least throughout the *Ascomycota* phylum. The species-specific differences are noted in the text.

## 2. Ergosterol-Enriched MCC

### 2.1. Subcellular Distribution of Ergosterol

Although their chemical structure slightly varies among the kingdoms, sterols are abundant in the membranes of most eukaryotic organisms. The flat, rigid molecules of these lipids modulate both membrane thickness and fluidity and are essential for the proper localization and function of various proteins [26,27,28]. The primary fungal sterol, ergosterol [29], is also important for proper function of the endocytic apparatus [30,31]. Ergosterol is not distributed equally among different membranes. About 70% of its total amount in yeast is localized in the plasma membrane, where it represents about 30–40 molar per cent of the lipid content [32,33]. Interestingly, almost 80% of plasma membrane ergosterol has been localized to its inner leaflet [34], i.e., to the sphingolipid-poor environment [35].

As revealed mostly by filipin staining, lateral distribution of ergosterol in the fungal membranes is not homogeneous either. Ergosterol is highly enriched in actively growing zones of the plasma membrane, including tips of mating projections (shmoo) in *S. cerevisiae*, *S. pombe* and *Cryptococcus neoformans*, ends of rod-shaped *S. pombe* cells, and tips of actively growing hyphae (but not inactive hyphae nor pseudohyphae) in *C. albicans* and *Aspergillus nidulans* [36,37,38,39,40]. Similarly, septa separating hyphal cells of *C. albicans* and *A. nidulans* and rod-shaped daughter cells of *S. pombe* are ergosterol-rich [37,39,41,42]. This lipid polarization is believed to be essential for hyphal morphogenesis and presentation of virulence factors [41,42], and proper localization of factors essential for membrane fusion during mating [36]. The elevated lipid accumulation at these sites is most-likely a steady-state manifestation of the targeted vesicular transport during the polarized growth (reviewed in [43]).

Besides the active growth zones, another ergosterol-rich domain of many fungal plasma membranes is the MCC [3]. MCC is home to several ergosterol-dependent nutrient transporters, such as Can1, Fur4 and Tat2, and its integrity is strongly affected in mutants defective in ergosterol biosynthesis [1,2,3,6].

Outside the plasma membrane, ergosterol-enriched domains have been described also in the vacuolar membrane of *S. cerevisiae* [44,45], which is somewhat surprising, as the vacuolar membrane had been considered an ergosterol-poor environment [29,46,47]. Available data strongly suggest that ergosterol-rich domains of the vacuolar membrane significantly increase in size after the diauxic shift of the yeast culture [44] and are involved in stationary phase lipophagy [45,48].

### 2.2. The Link between the MCC/Eisosome and Ergosterol Homeostasis

Disruption of ergosterol biosynthesis significantly affects the MCC integrity, structural uniformity and distribution of the microdomain in the plasma membrane, as reported by the Sur7 marker. Interestingly, the filipin stain pattern reporting the sterol enrichment of MCC is not completely disrupted in *erg2*Δ*, erg6*Δ and *erg24*Δ mutants [6], indicating that ergosterol precursors also accumulate in the MCC. However, the targeting of MCC-localized transporters to the plasma membrane is impaired under the conditions of defective ergosterol synthesis [1,49,50], suggesting that the accumulation of these transporters in the MCC is subordinate to ergosterol enrichment in the microdomain and not vice versa (Figure 1). Whether the change of MCC distribution and protein composition is a consequence of the unique properties of the ergosterol molecule missing in the plasma membrane, or the massive cellular adaptation to ergosterol absence leading to plasma membrane reorganization, remains unknown.

On the other hand, MCC integrity distinctly influences the ergosterol distribution within the plasma membrane. Deletion of the core eisosomal protein Pil1 completely prevents the MCC/eisosome formation in *S. cerevisiae* [8]. Filipin staining of these cells has revealed almost homogeneous distribution of ergosterol within the plasma membrane [3]. To a lesser extent, MCC/eisosome integrity is challenged in the absence of MCC proteins of the Nce102 family. Across *Ascomycota* phylum, numerous mutants lacking Nce102 homologues exhibit a reduced number of MCC microdomains [7,12,51,52]. In this case, ergosterol remains accumulated in the persisting microdomains [6], but their reduced number could lead to an increased ergosterol concentration in the rest of the plasma membrane of *nce102*Δ cells (Figure 1).

Upon inhibition of ergosterol biosynthesis by ketoconazole, increased expression of an Nce102 homologue Fhn1 [53], but not Nce102, has been reported in *S. cerevisiae* [54]. This change in Fhn1 expression appears to be regulated by the Upc2 and Ecm22 transcription factors [55] that positively regulate the expression of genes involved in sterol uptake and biosynthesis [56]. They also regulate expression of Npr1 [55], pointing to a possible link between nutrient-regulated TORC1 (Tor Complex 1) signalling and ergosterol biosynthesis. Upc2 responds directly to ergosterol abundance in the cytosol and, upon depletion of the lipid, relocalizes to the nucleus [57] where it activates the expression of several enzymes involved in ergosterol biosynthesis via binding to sterol regulatory elements (SRE) of their genes [56]. Interestingly, *FHN1* has been reported not to contain an SRE [54]. Furthermore, the rate of sterol uptake in *fhn1*Δ cells is indistinguishable from the wild type [55] and no direct interaction of Fhn1 with the ergosterol biosynthetic pathway has been reported to date. The underlying cause and function of increased Fhn1 levels following ketoconazole treatment therefore requires future attention.

In contrast to Fhn1, expression of Nce102 is independent of Upc2 and Ecm22 regulation [58]. Instead, it is negatively regulated by the Sut1 [59] and Sut2 [60] transcription factors that positively regulate sterol uptake genes in anaerobic conditions and are also involved in the regulation of filamentation [59]. Interestingly, while overexpression of either Nce102 or Fhn1 increases invasive growth of *S. ceevisiae* [58,61], providing a possible link between the MCC/eisosome and filamentous growth in *S. cerevisiae*, deletion of the respective genes (alone or in combination) has no detrimental effect on filamentous growth of the yeast [59]. On the other hand, the *nce102*Δ strain of *C. albicans* exhibits abnormal filamentous growth and a non-standard hyphal phenotype [51].

## 3. Eisosome and Sphingolipids

### 3.1. Fungal Sphingolipids

Sphingolipids, namely inositolphosphoceramides and their mannosylated derivatives, are another abundant class of fungal lipids [62,63]. In *S. cerevisiae*, they represent the majority of sugar-containing lipids throughout membranes [64]. In the plasma membrane specifically, they amount to 30% of the total phospholipids, i.e., about 7% of the plasma membrane mass [65]. Sphingolipids, especially those with very long acyl chains with a typical *all-trans* conformation [66], are essential for proper targeting of a plethora of proteins to the plasma membrane. Perhaps the most prominent examples are the H^+^-ATPase Pma1 [67,68] and the general amino acid permease Gap1 [69]. Being the dominant regulator of plasma membrane potential, Pma1 is one of the most abundant and most important proteins in the whole yeast cell [70,71]. Gap1, on the other hand, serves as a pivotal amino acid transporter when nitrogen sources are scarce [72].

Besides their indispensable structural function, fungal membrane lipids and the intermediates of their biosynthesis play a vital role in the regulation of a wide range of cellular processes. This is especially true for sphingolipids and their precursors (ceramides and long chain bases—dihydrosphingosine and phytosphingosine; LCBs) that play essential roles in the response to environmental stress conditions [73,74,75,76,77,78], response to ER stress [79,80], regulation of actin cytoskeleton polarization and endocytosis [64,81,82], calcium homeostasis, regulation of growth and the cell cycle, vesicle transport of glycosylphosphatidylinositol-anchored proteins, and many others (reviewed in detail in [83]). It is, therefore, quite clear that the biosynthesis of sphingolipids needs to be tightly regulated.

In general, little is known about the lateral distribution of sphingolipids within biological membranes. In the plasma membrane of fixed mouse fibroblasts, sphingolipid-enriched (but not cholesterol-rich) microdomains ~200 nm in diameter have been detected by secondary ion mass spectrometry [84,85]. While similar measurements have not been performed on fungal cells, time-resolved fluorescence spectroscopy uncovered the existence of highly ordered (gel-like) sphingolipid-enriched lipid microdomains in the plasma membrane of *S. cerevisiae* cells [86,87]. These membrane areas of unknown size and distribution within the yeast plasma membrane are supposed to buffer membrane order during fast stress adaptation [88,89].

There is no direct evidence estimating the relative abundance of sphingolipids within the MCC microdomain. However, the tightly packed gel-like character of sphingolipid-enriched microdomains described above makes them incompatible with the presence of the relatively bulky ergosterol molecules. As the MCC is considered a sterol-enriched plasma membrane microdomain (Section 2.1.), sphingolipid-enriched microdomains must be located outside of the MCC [89]. It is worthy to note here that also the aforementioned sphingolipid-dependent proteins Pma1 and Gap1 strictly avoid MCC localization [1,69,90]. Altogether, these facts do not necessarily mean that sphingolipids are excluded from the MCC, just that their relative abundance in the microdomain does not seem to be higher compared to the surrounding membrane. Phosphorylation of Pil1 and Lsp1 by another eisosome constituent, Pkh2, is strongly regulated by the presence of LCBs in vitro [10], which suggests that at least LCBs could be present inside the MCC.

### 3.2. Eisosomal Feedback Loop in Sphingolipid Biosynthesis

The first and rate-limiting step of sphingolipid biosynthesis in yeast is catalysed by the serine palmitoyltransferase complex (Lcb1-Lcb2-Tsc3; SPT) at the ER membrane [91,92]. Under the conditions of sphingolipid sufficiency, SPT activity is inhibited by Orm1 and Orm2 proteins (Figure 2a, Table 1), members of the *ORMDL* protein family that are highly conserved across species from unicellular fungi to humans [93]. The Orm proteins directly bind to both subunits of the catalytically active Lcb1-Lcb2 heterodimer [94,95], which causes inactivation of SPT via a currently unresolved molecular mechanism [96]. A decrease of sphingolipid levels results in reversible phosphorylation of the *N*-terminal domains of Orm proteins, leading to their dissociation from the SPT. The Orm proteins [91], as well as the SPT [96], then migrate to distinct locations within the ER membrane network. These changes ultimately result in activation of the SPT [91,96] and increase in the levels of long chain bases and their downstream products (Figure 2b).

Interestingly, the phosphorylation-mediated regulation of Orm1/2 binding to SPT is specific for fungi [91,93]. This regulation is executed by Ypk1 and Ypk2, the yeast homologues of mammalian serum-glucocorticoid kinases [91,97,98,99,100]. The kinase activity of Ypk proteins requires dual action of TORC2 (Tor Complex 2) complex and Pkh1/2 kinases, each phosphorylating Ypk1/2 at multiple condition-specific sites [97,98,101,102,103,104,105]. While Pkh1/2 accumulates at the eisosomes [106], TORC2 localizes to another microdomain at the plasma membrane [107].

A decade ago the sensing and regulation of sphingolipid levels was linked to the MCC/eisosome in *S. cerevisiae* [7] and several years later also in *A. nidulans* [52]. In their study, Fröhlich and co-workers observed the release of Nce102 from the MCC patches to the surrounding membrane and increased Pil1 phosphorylation following a decrease of sphingolipid levels [7] (Figure 2b). They suggested that MCC-accumulated Nce102 inhibits the activity of eisosome-localized serine-threonine kinases Pkh1/2 that are free to phosphorylate their natural target, Pil1 [10,106], only when Nce102 leaves the MCC microdomain. The underlying molecular mechanism of this Nce102-mediated regulation of Pkh activity, especially the act of sphingolipid sensing by the Nce102 molecule, has not been completely unravelled, however.

The consequences of Pkh activation seem to be better understood. The effect of Pil1 phosphorylation on eisosome stability is still a matter of some debate and seems to depend on the actual residues that are being phosphorylated [106,108,109]. However, it has been mentioned above (Section 2.2.) that the absence of the Nce102 protein reduces the eisosome number. Pil1 hyperphosphorylation was observed in *nce102*Δ cells, suggesting that Pkh activity resulting from the absence of Nce102 leads to eisosome disassembly. Accordingly, a decrease in the eisosome number was also detected in wild type cells after the inhibition of sphingolipid biosynthesis [7]. The eisosome disassembly releases the PI(4,5)P_2_-binding proteins Slm1/2 [11] that then diffuse laterally along the membrane to the TORC2 complex and activate it [22] (Figure 2B). The direct interaction of Slm1/2 proteins with TORC2 leads to their phosphorylation [110] and facilitates the binding of the serine-threonine kinases Ypk1/2 to TORC2, the first step of Ypk activation [111].

Recently, it has been suggested that Nce102 might be involved in this step of the pathway as well. Upon leaving the MCC, Nce102 physically interacts with Sng1 [112], a non-MCC integral plasma membrane protein [113], with which Nce102 appears to share some functional redundancy. While single deletions of either Nce102 or Sng1 do not display strong phenotypes, the double deletant exhibits severely impaired growth, higher sensitivity to oxidative stress and reduced chronological lifespan [112]. All of these phenotypes have been linked to the down-regulation of TORC2-Pkh/Ypk signalling [114,115,116]. Hence, at least one of Nce102 and Sng1 proteins needs to be expressed for proper response to sphingolipid status. Owing to their physical interaction and involvement in TORC2-Pkh/Ypk signalling, it has been suggested that Nce102 and Sng1 might act as a scaffold (that possibly also includes Slm1/2) that helps coordinate the localization and interaction among TORC2, Pkh kinases, and Ypk kinases [112], making the whole signalling module more efficient.

It has been recently reported that the expression of Ypk1 and Orm2 is increased in response to myriocin treatment [112]. Myriocin inhibits the SPT activity independently of Orm phosphorylation and thus impairs the Nce102-|Pkh1/2→Pil1→Slm1/2→TORC2→Ypk1-|Orm2-|SPT signalling pathway at its very end. This inhibition causes a drop in LCB levels and relocalization of Nce102 out of the MCC [7]. In the same line of thought, deletion of Nce102 also leads to elevated levels of Orm2 [112]. It is, therefore, possible that the presence of Nce102 in the MCC is an indirect regulator of Orm2 expression. Specifically, both myriocin-treated and *nce102*Δ cells have hyperphosphorylated Pil1 and severely affected eisosome stability [7], which ultimately leads to the activation of Pkh1, TORC2, and Ypk1/2, as described in more detail above. In turn, both TORC2 (via Slm1/2 [110,117,118]) and Ypk1/2 [111] negatively regulate the activity of the protein phosphatase calcineurin, a suppressor of *ORM2* expression [99,119,120]. The activation of TORC2 following myriocin treatment should, therefore, lead to a repression of calcineurin activity, resulting in the expression of genes under its control, including *ORM2*. Since the *nce102*Δ mutant has hyperphosphorylated Pil1 and destabilized eisosomes [7], creating an environment that should lead to SPT activation, the elevation of Orm2 proteins in these cells might be a part of cellular response to retain balance under conditions with inappropriately strong inputs from a potentially hyperactive Nce102-|Pkh1/2→Pil1→Slm1/2→TORC2→Ypk1-|Orm2-|SPT pathway.

The physiological relevance of Orm2 overexpression following myriocin treatment is somewhat less straight-forward to explain, since both Orm2 and myriocin act as inhibitors of SPT. Even more counter-intuitive (especially in the light of the preceding paragraph) is the fact that myriocin treatment also induces expression of Nce102 [112,121]. We propose that building a more robust regulatory apparatus when sphingolipid demand increases could, in fact, represent the rescaling of the whole system to a higher sensitivity and/or dynamic, enabling both rapid elevation and decrease of sphingolipid biosynthesis upon the action of a wide range of stimuli. A good representative example of this is the cellular response to heat stress. Namely, the adaptation to elevated temperatures necessarily includes an increase of the sphingolipid content in the plasma membrane [122,123]. However, excessive sphingolipid levels would cause the membrane to become too rigid, a scenario incompatible with life.

To understand how the eisosome is re-assembled when the activation of sphingolipid biosynthesis brings the desired effect, let us consider the origin of Pil1 denomination, i.e., Phosphorylation inhibited by long chain bases [10]. Although the Pil1 name was assigned based on in vitro experiments, it seems clear that products of the sphingolipid synthesis mediate a direct feedback loop to regulate the eisosome integrity also in vivo. Supplementing SPT-compromised cells with long chain bases for as short as 15 min resulted in the de novo formation of MCC/eisosomes. In addition, the accumulation of complex sphingolipids in the absence of the inositol phosphosphingolipid phospholipase C Isc1 even caused the hyperassembly of Pil1 eisosomes [7]. However, our knowledge regarding this feedback mechanism is far from complete. For example, the identity of a phosphatase responsible for Pil1 dephosphorylation in this scenario remains to be uncovered. Furthermore, the exact role of the Pil1 homologue Lsp1 in the eisosome assembly remains a mystery, as Lsp1 is by itself not sufficient for the eisosome to form in *pil1*Δ cells under normal conditions. Furthermore, while the LCBs inhibit the phosphorylation of Pil1, they promote the phosphorylation of Lsp1 (Lsp1—Long chain bases stimulate phosphorylation), suggesting the opposite behaviour of Lsp1 upon sphingolipid depletion.

### 3.3. Further Notes on Nce102′s Function as a Sphingolipid Sensor

As mentioned above, the molecular details of changes in the Nce102 molecule, which promote its release from the MCC upon sphingolipid depletion, have not been identified. However, behaviour similar to that of Nce102 has been described for some other protein constituents of the MCC/eisosome. A conserved 5′-3′ exoribonuclease Xrn1 reversibly associates with eisosomes only after the gradual depletion of fermentable carbon sources [23] to attenuate its exoribonuclease activity [124], directly linking the eisosome with the regulation of mRNA turnover. Specific nutrient transporters accumulate within the MCC in the absence of their substrates [1,2,3,4,5] to prevent untimely degradation [6,125]. While little is known about Xrn1 binding to the eisosome, the mechanism of the lateral redistribution of MCC transporters has been described in detail recently.

Regarding the example of the arginine permease Can1, Gournas and co-workers demonstrated that the transport-elicited release of transporters from the MCC microdomain is caused by a specific conformational change in the protein molecule upon substrate binding [125]. The authors suggested that due to the asymmetric distribution of charged residues on both sides of the membrane, a similar change of Can1 conformation could also be expected in response to variations of transmembrane potential. This would explain why Can1 was released from MCC to the surrounding membrane in response to partial or total membrane depolarization, independent of the cause of membrane potential decrease (i.e., transport- or ionophore-induced uncoupling of the proton gradient, membrane permeabilization by external electric field or respiration insufficiency) [3]. Similar to Nce102, Can1 also accumulated in the MCC only under conditions of normal sphingolipid biosynthesis [125]. In contrast to Can1, however, Nce102 did not respond to changes in membrane potential [6], indicating a difference between the mechanisms regulating Nce102 and Can1 accumulation in the MCC.

It should be mentioned in this context that the truncation of the charged C terminus of the Nce102 molecule prevents accumulation of both Nce102 and Can1 in the MCC [53]. Truncated Nce102, thus, resigns its function as a sphingolipid sensor. However, since the C terminus of Nce102 has been localized to the cytoplasm, its direct interaction with sphingolipids is not plausible. Instead, this truncation could shift the balance between different conformations of the Nce102 molecule towards the one induced by decreased sphingolipid levels. The suggested flexible hairpin structure [53] makes the Nce102 molecule ideally predisposed to switch between different conformations—one for the localization in the curved, ergosterol-enriched membrane of the MCC microdomain and the other for travelling away to the flat parts of the plasma membrane. The engagement of the C terminus in triggering the respective Nce102 conformations remains enigmatic.

## 4. Plasma Membrane Phosphatidylinositols and MCC

Glycerophospholipids are the most abundant constituents of biological membranes. Their polar head-groups and hydrophobic carbon chains give them their amphiphilic character, which enables the spontaneous formation of lipid bilayers, the platforms of life. Phosphoinositides represent only about 1% of total cellular phospholipids [126,127]. Despite their relatively low abundance, they are important regulators of various aspects of endocytosis (reviewed in detail in [128]), cell growth, gene transcription, and many other processes [129,130,131,132,133]. Various forms of phosphoinositides act as second messengers in a wide range of signalling pathways, a function mediated by their reversible phosphorylation by a set of kinases and lipases [134].

The most abundant phosphoinositides in yeast are PI(4,5)P_2_ (phosphatidylinositol 4,5-bisphosphate) and its precursor, PI(4)P. Both are essential components of the plasma membrane [135], where they are synthesised in situ. First, PI(4)P is produced by the phosphorylation of phosphatidylinositol by the Stt4 kinase [136,137], which is followed by Mss4-mediated phosphorylation of PI(4)P to produce PI(4,5)P_2_ [138,139].

As revealed by fluorescence microscopy, phosphoinositides are distributed mostly homogeneously throughout the inner leaflet of the plasma membrane [140,141,142,143], although functional exceptions do exist. In *C. albicans*, PI(4,5)P_2_ is enriched at the tip of both buds and actively growing hyphae. This polarization of the lipid is necessary for the proper growth of both [144]. In fact, sufficient PI(4,5)P_2_ levels possibly represent a critical factor for the transition of *C. albicans* from budding to filamentation [145].

In contrast to the lipids themselves, the kinases and phosphatases regulating phosphoinositide phosphorylation are generally concentrated into specialized membrane domains [146,147]. The MCC/eisosome is one such domain. The eisosome accumulates several proteins that directly bind PI(4,5)P_2_, either via their BAR domains (Pil1, Lsp1), pleckstrin homology domains (Pkh2), or both (Slm1/2) [6,11,14,22,148]. In fact, assembly of the eisosome itself is dependent on sufficient PI(4,5)P_2_ levels in the plasma membrane of both *S. cerevisiae* and *S. pombe*. Similarly, in vitro, Pil1 and Lsp1 tubulate PI(4,5)P_2_ containing membranes [13,14,149]. On the other hand, the *S. cerevisiae* eisosome also recruits the cytosolic PI(4,5)P_2_ phosphatase Inp51/Sjl1 via Pil1 that is indispensable for the interaction [21,150]. Interestingly, the *S. pombe* homologue of Inp51, Syj1, does not exhibit a strong colocalization with Pil1, but both proteins have been shown to be part of the same PI(4,5)P_2_ regulatory pathway, regardless [14].

The deletion of Pil1 in *S. cerevisiae* decreases the interaction of Inp51/Sjl1 with the plasma membrane, resulting in a 1.3-fold increase in the total cellular level of PI(4,5)P_2_ [21,150]. The amount of PI(4,5)P_2_ available for protein binding increases by an even greater factor in the *pil1*Δ deletion strain [21], which is most likely due to the absence of an eisosome structure in these cells, making the PI(4,5)P_2_ more accessible. Taken together, these data indicate that the MCC/eisosome plays an important role in the regulation of PI(4,5)P_2_ level in the plasma membrane. This regulation is mediated by concentrating the lipid in the MCC by eisosomal proteins, whereupon it is dephosphorylated by Inp51/Sjl1.

The importance of this regulation is demonstrated by the phenotype displayed by *C. albicans inp51*Δ and *irs4*Δ mutants (Irs4 binds Inp51 and is required for its activation). These cells exhibit abnormal cell wall morphogenesis forming tubular cell wall invaginations [151], similar to those formed in *sur7*Δ and *pil1*Δ*lsp1*Δ mutants [17,152]. Similarly, the *S. cerevisiae pil1*Δ mutant, which is defective in localizing the Inp51/Sjl1 to the plasma membrane [150], displays cell wall irregularities [8].

Interestingly, osmotic stress induces formation of PI(4,5)P_2_ clusters in the plasma membrane of *S. pombe* cells that also accumulate the PI(4)P-5 kinase Its3. It is, therefore, possible that PI(4,5)P_2_ is synthesized in these clusters from PI(4)P, resulting in their elevated local concentration. While these clusters are distinct from MCC/eisosomes, they are spatially organized by them in both fission and budding yeast cells [153].

## 5. Functional Relevance of MCC/Eisosome-Mediated Lipid Regulation

Although discussed separately above, the biosynthetic and regulatory pathways of fungal sterols, sphingolipids, and phosphoinositides are tightly interconnected [20,92,99,118,154,155,156,157,158,159,160,161,162,163,164]. This makes the eisosome a part of a wide network tightly controlling the cellular lipidome. In this network, causality is often masked and predictions can be extremely difficult. Mutual relations turn in circles, in which the contribution of the players is not easy to quantify.

As an example of this complexity, sterols are transported from the ER to the plasma membrane (PM) at the ER-PM contact sites by the oxysterol binding protein Osh4 in exchange for PI(4)P [165,166,167,168]. PI(4)P levels in the ER membrane are reduced by the phosphatase Sac1 catalysing the reduction of PI(4)P to phosphatidylinositol [169,170], a lipid essential for the conversion of ceramide to inositol phosphoceramide (IPC) in later stages of sphingolipid biosynthesis [162]. This could be the reason why the deletion of *SAC1* leads to an accumulation of LCBs and a decrease in complex sphingolipids in *sac1*Δ cells [162]. In addition, the direct binding of Sac1 to the Orm1/2-SPT complex has also been observed [91,96]. To complete the circle, the positioning of the cortical ER (and therefore also the ER-PM contact sites) is regulated by the sphingolipid-dependent MCC/eisosomes [171].

Recently, a direct connection between TORC2 activation and sterol distribution in the plasma membrane has been revealed. It has been shown that the retrograde plasma membrane-to-ER transport of sterols, mediated at PM-ER contact sites by the protein Ysp2/Lam2/Ltc4 (and to a lesser degree by its paralogue, Lam4/Ltc3) is blocked following their phosphorylation by Ypk1 [172]. Among other things, this means that the disintegration of MCC/eisosomes following the reduction of sphingolipid levels, which leads to TORC2 activation, results not only in the launching of sphingolipid biosynthesis, but also in preventing the losses of plasma membrane ergosterol. Both of these processes are mediated by the action of TORC2-activated Ypk1 kinase.

The functional relevance of such connections becomes visible upon system rearrangements—during metabolic switches between fermentation and respiration or during the adaptation to various types of environmental stress. It seems just a natural consequence of the presented feedback loop between the eisosome assembly/disassembly and sphingolipid levels (Figure 2) that deletion of the *PIL1* gene confers increased thermotolerance [10]. In *pil1*Δ cells, eisosomes cannot be assembled [8], which, according to the model described in Section 3.2., triggers SPT activation. The extent of the overall difference between plasma membrane lipidomes in *pil1*Δ and wild type cells is currently unknown.

Translocation of Slm proteins from the MCC towards TORC2 can be induced not only by a decrease in sphingolipid levels, but also by mechanically induced plasma membrane deformation [22]. Apparently, the excess membrane within the MCC provides the cell not only with extended surface elasticity [173], but also with a sensitive, PI(4,5)P_2_ dependent, signalling of mechanical stress. Again, the membrane distortion is followed by the mobilization of lipid synthesis to reset the normal membrane tension [174] (Figure 2). Similarly, upon cell shrinkage during dehydration, the MCC furrows are capable of deepening inside the cell and partially absorbing the surplus membrane, which stays there, ready to be used in the case of eventual rehydration [25].

Little is known about the involvement of the MCC/eisosome in response to oxidative stress. However, the eisosomal binding of three flavodoxin-like proteins (Pst2, Rfs1, and Ycp4) upregulated during the oxidative stress has been observed [6,175]. Furthermore, a massive rearrangement of the plasma membrane, including the adaptation of the phospholipid composition, as well as the shape and distribution of MCC/eisosomes, was detected during the cell adaptation to H_2_O_2_ exposure [24]. Differential susceptibility of MCC/eisosome mutants to various oxidative stress generators in *S. cerevisiae* [176] and *A. nidulans* [52] indicated direct functional involvement of the microdomain in regulation of the cellular response to oxidative stress, probably via modulation of Ypk-mediated signalling.

## 6. Concluding Remarks and Future Prospects

To date, about one hundred studies have been published concerning the fungal plasma membrane microdomain MCC/eisosome. However, there are still many “details” we do not understand. Although we do not consider the known proteome of the MCC/eisosome to represent the complete set, we do not expect any major discoveries in this respect in the near future. The actual imperatives concerning the MCC seem to be different.
**Lipid composition of the MCC:** As stated above, there are valid reasons to expect (sphingolipid-free) ergosterol, as well as some PI(4,5)P_2_, inside the MCC membrane. The question of the presence of sphingolipids in the microdomain is more complicated. The MCC membrane is detached from the cell wall, indicating a low concentration of glycosylated lipids. However, this does not exclude the presence of non-glycosylated sphingolipid precursors, namely LCBs and ceramides, within the compartment. The limited possibilities of specific labelling of lipid species in vivo accent the necessity of biochemical characterization of the microdomain lipid composition. However, all previous attempts to isolate the MCC/eisosome have failed.**Mechanism of Nce102 action**: One of the biggest gaps in the understanding of the MCC/eisosome-mediated sphingolipid biosynthesis regulation is the understanding of the molecular mechanism of Nce102 redistribution out of the MCC upon sphingolipid depletion. In this respect, *S. cerevisiae* represents a good model, as it contains exactly two Nce102-like proteins in the genome. In addition, Nce102 and Fhn1 expression profiles significantly differ with respect to their dependence on nutrient availability and actual lipid biosynthetic activity. For example, Fhn1 expression is induced by Upc2 following sterol biosynthesis inhibition, while Nce102 is not [54,55,58]. Whether this is just a reflection of the tight interconnection between the ergosterol and sphingolipid metabolic pathways, or whether Fhn1 plays an unknown sphingolipid-independent function in the membrane remains unknown. Even in the case of the proposed sphingolipid sensor Nce102, no direct experimental evidence has excluded the possibility that ergosterol, not sphingolipids, could interact with the Nce102 molecule. The act of sphingolipid sensing could in fact be the sensing of ergosterol-sphingolipid imbalance in the membrane.**Cell interior and MCC/eisosome:** The possible involvement of the MCC/eisosome in ergosterol metabolism opens other directions for future studies. For example, the eisosome function has not been related to the architecture of the inner cellular membranes, although the organelle morphology and membrane lipid composition are closely related. If involved in ergosterol biosynthesis and regulation, the MCC/eisosome could influence morphology of various cellular organelles. The first evidence in support of a role for eisosomes in autophagy was reported in a recent study focused on the pathogenic fungus, *Beauveria bassiana* [177].

Although far from being comprehensive, the above list clearly illustrates that while our knowledge of the MCC/eisosome function in the regulation of lipid homeostasis is quite extensive, there are still long ways to travel and mysteries to unravel before we fully understand it in a cellular context.

## Figures and Tables

**Figure 1 biomolecules-09-00305-f001:**
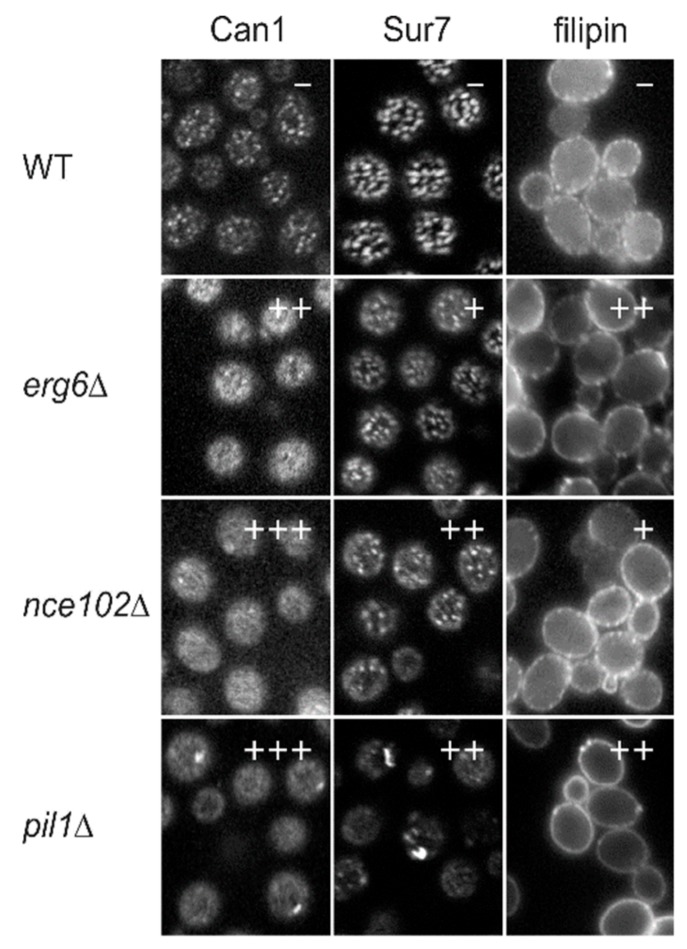
Distribution of membrane compartment of Can1 (MCC) markers in selected knockout strains. Distributions of Can1-GFP, Sur7-GFP, and filipin-stained sterols were monitored in the library of single gene deletion strains. Examples of detected phenotypes (classification of phenotypes: wild type [WT]—like, −; weak, +; medium, ++; strong, +++) on tangential confocal sections (Can1 and Sur7) or wide-field images (filipin; transversal sections) are presented. Figure adapted from [6] (©2008 Grossmann et al. Originally published in Journal of Cell Biology. https://doi.org/10.1083/jcb.200806035).

**Figure 2 biomolecules-09-00305-f002:**
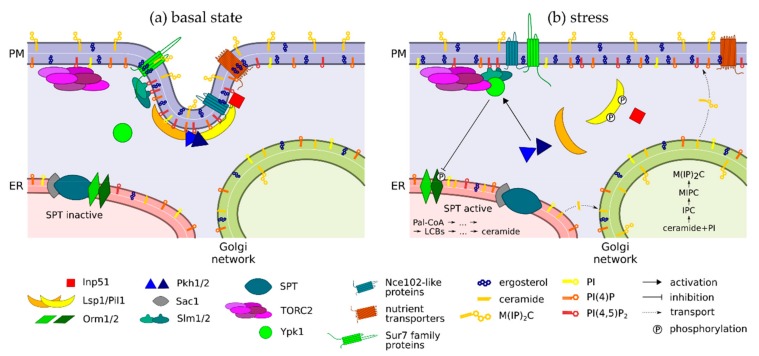
Interplay of MCC/eisosomes and sphingolipid biosynthesis. (**a**) The MCC/eisosome is fully assembled under basal, non-stress conditions. (**b**) MCC/eisosome disassembles following depletion of sphingolipids and/or mechanical stretching of the plasma membrane, resulting in activation of TORC2 and in turn SPT, catalysing the first step of sphingolipid biosynthesis. Short descriptions of proteins involved in this model are listed in Table 1. Used abbreviations: ER—endoplasmic reticulum; IPC—inositol phosphoceramide; LCBs—long chain bases; MCC—membrane compartment of Can1; M(IP)_2_C—mannose-(inositol-P)_2_-ceramide; MIPC—mannosyl-inositol phosphoceramide; Pal-CoA—palmitoyl coenzyme-A; PI—phosphatidylinositol; PI(4,5)P_2_—phosphatidylinositol 4,5-bisphosphate; PI(4)P—phosphatidylinositol 4-phosphate; PM—plasma membrane; SPT—serine palmitoyltransferase; TORC2—Tor Complex 2.

**Table 1 biomolecules-09-00305-t001:** Proteins involved in the MCC/eisosome-mediated regulation of sphingolipid metabolism (see also Figure 2).

**MCC/Eisosome Components**	
Lsp1/Pil1	- BAR (*Bin/Amphiphysin/Rvs*) domain proteins—bind PI(4,5)P_2_ at the plasma membrane - Core eisosome constituents, essential for eisosome formation (Pil1) - Differentially phosphorylated by Pkh1/2 in response to LCB levels
Nce102-like -proteins	- Tetraspan proteins -Anchor nutrient transporters to MCC - Change their plasma membrane distribution in response to sphingolipid content (Nce102)
nutrient -transporters	- APC (amino acid polyamine organocation) transporters localizing to MCC - Plasma membrane trafficking of most of these was shown to be ergosterol-dependent
Slm1/2	- BAR and pleckstrin homology domain proteins—bind PI(4,5)P_2_ at the plasma membrane - Travel between MCC and TORC2 (Tor Complex 2) in response to membrane stress - Activate TORC2
**Kinases**	
Pkh1/2	- Sphingolipid-dependent kinases - Inhibited by Nce102 at eisosomes
TORC2	- Tor Complex 2, composed of six known proteins (including Avo1-3) - Associated to plasma membrane, likely via pleckstrin homology domain protein Avo1 - Activated during membrane stress by Slm1/2
Ypk1	- Inhibitor of Orm2, i.e., indirect activator of SPT (serine palmitoyltransferase) - Dually activated by Pkh1/2 and TORC2
**Phosphatases**	
Inp51	- PI(4,5)P_2_ phosphatase - Recruited to eisosome via interaction with Pil1
Sac1	- PI(4)P phosphatase - Forms a higher-order complex with SPT
**Others**	
Orm1/2	- Inhibitors of SPT - Inactivated by Ypk1-mediated phosphorylation (Orm2)
SPT	- Serine palmitoyltransferase complex, composed of Lcb1, Lcb2 and Tsc3 - Catalyses the first and rate-limiting step of sphingolipid biosynthesis

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
