# Peer review of "Role of MCC/Eisosome in Fungal Lipid Homeostasis"

_biomolecules, 2019, doi:10.3390/biom9080305_

Round 1
Reviewer 1 Report
This is an apparently well-written review of the relationships between eisosomes and lipids, material that I am not familiar with. Therefore, I am unqualified to judge its accuracy, completeness, or conceptualizations. Assuming that the authors have accurately summarized the content of the literature, it is publishable and may be of interest for readers with expertise in membrane proteomics and metabolism.
However, I find the detailed discussion of dozens of components and hundreds of interactions impossible to assimilate. Adding illustrations for known and proposed pathways, protein-protein interactions, protein-lipid interactions and signaling would be a big help.
Specific points:
Line 19: RNA decay is mentioned nowhere else in the MS, and so does not belong in the Abstract
l. 55: “Specific shape and random orientation…” The meaning is not clear, in that a mechanism of sensing and response is not yet described in the text. Perhaps a sentence or two about the results in ref [22] would help.
l. 105: A figure showing the relationship of ergosterol biogenesis with MCC/eisosomes would be helpful.
l. 106-110: “the presence of ergosterol-dependent transporters in MCC is not the primary driving force…” The logic of this conclusion is not clear.
l. 122-125: Which species is being described here?
l. 135-138: The previous paragraph describes FHN1 as non-functional. Therefore, describing it here as a “functional homolog” is confusing.
l. 265: “ascertain” may be a wrong word. Is the intended meaning that LCB’s are part of a pathway that regulated MCC assembly? Again here an illustration or schema would greatly help the reader.
l. 315: What is the antecedent of “their”? What is being phosphorylated?
Section 5 is extremely difficult to understand. A pathway schema would be extremely helpful.
Author Response
Response to comments of Reviewer #1
This is an apparently well-written review of the relationships between eisosomes and lipids, material that I am not familiar with. Therefore, I am unqualified to judge its accuracy, completeness, or conceptualizations. Assuming that the authors have accurately summarized the content of the literature, it is publishable and may be of interest for readers with expertise in membrane proteomics and metabolism. However, I find the detailed discussion of dozens of components and hundreds of interactions impossible to assimilate – Adding illustrations for known and proposed pathways, protein-protein interactions, protein-lipid interactions and signaling would be a big help.
Specific points:
Line 19: RNA decay is mentioned nowhere else in the MS, and so does not belong in the Abstract
Response: The involvement of the eisosome in RNA decay is mentioned in section 3.3. We have adjusted the respective text in the MS slightly to highlight this connection.
55: “Specific shape and random orientation…” The meaning is not clear, in that a mechanism of sensing and response is not yet described in the text. Perhaps a sentence or two about the results in ref [22] would help.
Response: We re-formulated this part of the text to make it more digestible.
105: A figure showing the relationship of ergosterol biogenesis with MCC/eisosomes would be helpful.
Response: As mentioned in the text, molecular details concerning the mutual interference of the ergosterol biosynthesis and the MCC/eisosome remain unknown. While we collected the main indices pointing to the existence of this interconnection in section 2.2., we are afraid that the data published so far are not sufficient to draw an unequivocal causal scheme here. However, we included a figure adapted from Grossmann et al., 2008, demonstrating both the alteration of MCC morphology in the mutant defective in ergosterol biosynthesis, as well as changes in localization of the filipin stain in mutants defective in MCC/eisosome formation.
106-110: “the presence of ergosterol-dependent transporters in MCC is not the primary driving force…” The logic of this conclusion is not clear.
Response: The whole paragraph has been re-arranged in the revised version of the manuscript. We avoided the problematic sentence.
122-125: Which species is being described here?
Response: It is S. cerevisiae. We added this specification to the text.
135-138: The previous paragraph describes FHN1 as non-functional. Therefore, describing it here as a “functional homolog” is confusing.
Response: To avoid any confusion of the reader, we deleted “functional homolog” from the text. The Reviewer was right, we do not need it there. Just to explain to the Reviewer: “functional homolog of Nce102” is the name description for Fhn1. The name refers to the ability of Fhn1 to accumulate itself and MCC-specific transporters to the microdomain, similar to Nce102.
265: “ascertain” may be a wrong word. Is the intended meaning that LCB’s are part of a pathway that regulated MCC assembly? Again here an illustration or schema would greatly help the reader.
Response: Exchanged as suggested.
315: What is the antecedent of “their”? What is being phosphorylated?
Response: We specified this in the revised version.
Section 5 is extremely difficult to understand. A pathway schema would be extremely helpful.
Response: Unfortunately, there is no comprehensive scheme illustrating the interplay among metabolic pathways of different lipid species. In fact, there are too many already known connections to be included in one diagram and even these are far from being complete. Concerning the involvement of MCC/eisosome, we know solid molecular details only for sphingolipid metabolism and these are summarized in Figure 2 and Table 1 in the revised version of the manuscript. We accented in the text that the introductory story presented in the second paragraph of the Section is just an example of possible functional interconnections. We also added one more paragraph to Section 5, illustrating the direct involvement of the eisosome in ergosterol metabolism. We decided however not to include the Figure into this section.
Reviewer 2 Report
The authors in this mini-review are focused on the role of MCC/eisosome, a detailed-studied fungal plasma membrane microdomain, in the regulation of lipid metabolism and on the molecular mechanism(s) underlying this intracellular process in Saccharomyces cerevisiae.
Major points
1. The authors suggest that due to the asymmetric distribution of charged residues on both sides of the membrane, similar change of Can1 conformation could be expected also in response to variation of the transmembrane potential.However, in a recent article (Bianchi et al., 2019) although it was confirmed that Can1 was moved away from the MCC/eisosomes domain in the presence of its substrate arginine, similar localization patterns for Can1 were observed in the presence and absence of FCCP (a protonophore dissipated the (electro) chemical proton gradient), suggested that unlike the substrate, the proton-motive force appears to play little or no role in the PM distribution of Can1. This is missing in the submitted mini-review. Please add this information in the revised manuscript and comment on that.
2. It is well documented that the initial idea of differential segregation of lipids being the forming mechanism of lipid rafts suggested the existence of lipid-driven membrane domains. However, several mechanisms of protein-driven membrane domains are also well documented. Such protein-driven domains, among others, are those organized by specialized scaffolding proteins, as is the case of eisosomes in fungi and caveolae in mammalian cells. To my knowledge, from the above studies it was proposed that, it is more likely that small, lipid-driven nanoscale domains are transiently formed and mainly consist of lipids (e.g. lipid rafts), while more stable domains are usually larger and their organization and composition is mainly dictated by proteins (Carquin et al., 2016, Kusumi et al., 2011).
To briefly mention and comment on the above is necessary in this mini-review article.
3. Regarding the involvement of the MCC/eisosome in response to the oxidative stress it has been shown that different superoxide generators result in differential effects on growth of strains lacking the genes encoding the PilA and Nce102 eisosomal proteins (Athanasopoulos et al., 2015). These results suggested a mechanism other than superoxide formation being responsible for this discrepancy. This information might be necessary in the revised manuscript.
4. Page 373: A diagram will be helpful.
Minor point
Page 210: unraveled. Please correct.
Author Response
Response to comments of Reviewer #2
The authors in this mini-review are focused on the role of MCC/eisosome, a detailed-studied fungal plasma membrane microdomain, in the regulation of lipid metabolism and on the molecular mechanism(s) underlying this intracellular process in Saccharomyces cerevisiae.
Major points 1.
The authors suggest that due to the asymmetric distribution of charged residues on both sides of the membrane, similar change of Can1 conformation could be expected also in response to variation of the transmembrane potential. However, in a recent article (Bianchi et al., 2019) although it was confirmed that Can1 was moved away from the MCC/eisosomes domain in the presence of its substrate arginine, similar localization patterns for Can1 were observed in the presence and absence of FCCP (a protonophore dissipated the (electro) chemical proton gradient), suggested that unlike the substrate, the proton-motive force appears to play little or no role in the PM distribution of Can1. This is missing in the submitted mini-review. Please add this information in the revised manuscript and comment on that.
Response: We are indeed aware of the study of Bianchi et al. (2018); we cite it elsewhere in the text of the manuscript. However, this study was focused on a slightly different topic. It presented an excellent and precise localization and mobility tracking of several MCC/eisosome components, for example. The membrane depolarization by FCCP was showed there just in one of the supplementary figures, and unfortunately, it was done without a precise specification of the used experimental procedure. In the Methods, the authors said that the cells were suspended in a fresh (?) medium and then immobilized “in between two microscope slides“ (in the case of FRAP analyses), or “embedded in 0.5% (w/v) low-melting agarose and placed in between two microscope slides“ (in the case of size determination of the MCC/eisosomes). It is obvious that an FCCP treatment is capable to induce the proton influx only in acidic media, when the excess protons outside the cell passively run through the ionophore-permeabilized membrane along the concentration gradient (discussed in [3]). In this experiment, however, we do not know anything concerning the extracellular pH. Therefore, we cannot judge the relevance of the presented results. Only for this reason we decided not to include the FCCP-related data from Bianchi et al., 2018 to the argumentation commented by the Reviewer.
2. It is well documented that the initial idea of differential segregation of lipids being the forming mechanism of lipid rafts suggested the existence of lipid-driven membrane domains. However, several mechanisms of protein-driven membrane domains are also well documented. Such protein-driven domains, among others, are those organized by specialized scaffolding proteins, as is the case of eisosomes in fungi and caveolae in mammalian cells. To my knowledge, from the above studies it was proposed that, it is more likely that small, lipid-driven nanoscale domains are transiently formed and mainly consist of lipids (e.g. lipid rafts), while more stable domains are usually larger and their organization and composition is mainly dictated by proteins (Carquin et al., 2016, Kusumi et al., 2011). To briefly mention and comment on the above is necessary in this mini-review article.
Response: The Reviewer is right that various mechanisms of membrane microdomain formation coexist in the living cell. We strongly disagree however that the suggested simple generalization, saying that there are only two types of membrane microdomains, when lipid-driven microdomains are small (nanoscale) and transient while the protein-driven ones are large and stable, can be drawn.
In our minireview, we mention, for example, large, ergosterol-rich microdomains, which are not supported by any protein scaffold, but persist as micron-sized, steady-state plasma membrane areas in growing cells for periods comparable to a cell cycle duration, and their existence is conditioned by the ongoing vesicular transport (reviewed in [43], for example). Similarly, we mention stable, ergosterol-enriched microdomains in a vacuolar membrane of stationary-phase cells which are probably a result of lipid phase separation. These microdomains have been in extenso discussed elsewhere, for example in Malinsky and Opekarova, 2016 (Malinsky J and Opekarova M, New Insight Into the Roles of Membrane Microdomains in Physiological Activities of Fungal Cells. In: Kwang Jeon, International Review of Cell and Molecular Biology 325:119-180, ISBN:978-0-12-804806-1, Elsevier Inc. Academic Press, 2016). Finally, the stable, highly ordered, sphingolipidenriched microdomains in the yeast plasma membrane mentioned in Chapter 3 of our manuscript are another example of lipid-driven microdomains which do not fit this binary classification.
We are of the opinion that a broader discussion of multiplex mechanisms leading to membrane microdomain formation strongly exceeds the scope of the presented minireview, which is strictly focused on one single (protein-driven, as there is no MCC without Pil1) plasma membrane microdomain, and we have not included that into the revised version of the manuscript.
3. Regarding the involvement of the MCC/eisosome in response to the oxidative stress it has been shown that different superoxide generators result in differential effects on growth of strains lacking the genes encoding the PilA and Nce102 eisosomal proteins (Athanasopoulos et al., 2015). These results suggested a mechanism other than superoxide formation being responsible for this discrepancy. This information might be necessary in the revised manuscript.
Response: We are grateful for this comment. We extended the relevant part of the text in the revised version of the manuscript.
4. Page 373: A diagram will be helpful.
Response: Unfortunately, there is no comprehensive scheme illustrating the interplay among metabolic pathways of different lipid species. In fact, there are too many already known connections to be included in one diagram and even these are far from being complete. Concerning the involvement of MCC/eisosome, we know solid molecular details only for sphingolipid metabolism and these are summarized in Figure 2 and Table 1 in the revised version of the manuscript. We accented in the text that the introductory story presented in the second paragraph of the Section is just an example of possible functional interconnections. We also added one more paragraph to Section 5, illustrating the direct involvement of the eisosome in ergosterol metabolism. We decided however not to include the Figure into this section.
Minor point Page 210: unraveled. Please correct.
Response: The British English was used throughout the manuscript. In this, “unravelled” is a correct spelling; “unraveled” is American.
Reviewer 3 Report
Article review on “ Role of MCC/eisosome in fungal lipid homeostasis”
The authors provided a good introduction for each section of the manuscript and covered the latest knowledge well on the relationship between MCC/eisosome and lipids dynamics. Overall, it is well written and will attract many readers in the field. The manuscript does not, however, contain a single schematic diagram that could promote readers’ understanding on the written topics. In sum, the reviewer suggests the journal to accept the manuscript with a revision of the current manuscript with the suggested changes below.
Specific points by the reviewer:
Line 51: to localize a sensory organ there as wellè not clear if the organ denotes lipid homeostasis sensory apparatus?
Line 80: 30-40 molar per cent of the lipid content è any particular reason to separate two words “per” and “cent”?
Schematic diagrams: in the section of “the link between the MSS/eisosome and ergosterol homeostasis”
Molecular interaction or action model diagram for section 3-2 in particular lines from 211-221.
Lines 225-226: While single deletion strains do not display strong phenotypes, the double deletant has severelyimpaired growth, higher sensitivity to oxidative stress and reduced chronological lifespanè clearly indicate double deletion of Nce102 and Sng1.
Line 341: The amount of PI(4,5)P2 available for protein binding increases by an even greater factor in the pil1Δ deletion strain. This discrepancy is most likely due toè not sure if this is discrepant. Since the author clearly explained that loss of Pil1 results in loss of eisosomes, but PIP2 level increases in this condition. Not a major issue, but a delicate change in choosing words is required here.
Author Response
Response to comments of Reviewer #3
The authors provided a good introduction for each section of the manuscript and covered the latest knowledge well on the relationship between MCC/eisosome and lipids dynamics. Overall, it is well written and will attract many readers in the field. The manuscript does not, however, contain a single schematic diagram that could promote readers’ understanding on the written topics. In sum, the reviewer suggests the journal to accept the manuscript with a revision of the current manuscript with the suggested changes below.
Specific points by the reviewer:
Line 51: to localize a sensory organ there as wellè not clear if the organ denotes lipid homeostasis sensory apparatus?
Response: We specified this in the revised version of the manuscript.
Line 80: 30-40 molar per cent of the lipid content è any particular reason to separate two words “per” and “cent”?
Response: Just a personal British English spelling preference highlighting the actual meaning of “in a hundred”.
Schematic diagrams: in the section of “the link between the MSS/eisosome and ergosterol homeostasis”
Response: As mentioned in the text, molecular details concerning the mutual interference of the ergosterol biosynthesis and the MCC/eisosome remain unknown. While we collected the main indices pointing to the existence of this interconnection in section 2.2., we are afraid that the data published so far are not sufficient to draw an unequivocal causal scheme here. However, we included a figure adapted from Grossmann et al., 2008, demonstrating both the alteration of MCC morphology in the mutant defective in ergosterol biosynthesis, as well as changes in localization of the filipin stain in mutants defective in MCC/eisosome formation.
Molecular interaction or action model diagram for section 3-2 in particular lines from 211-221.
Response: We added the diagram to section 3.2. as suggested by the Reviewer.
Lines 225-226: While single deletion strains do not display strong phenotypes, the double deletant has severely impaired growth, higher sensitivity to oxidative stress and reduced chronological lifespanè clearly indicate double deletion of Nce102 and Sng1.
Response: This has been clarified in the text.
Line 341: The amount of PI(4,5)P2 available for protein binding increases by an even greater factor in the pil1Δ deletion strain. This discrepancy is most likely due toè not sure if this is discrepant. Since the author clearly explained that loss of Pil1 results in loss of eisosomes, but PIP2 level increases in this condition. Not a major issue, but a delicate change in choosing words is required here.
Response: The sentence has been rephrased in the revised version of the manuscript. We have avoided “discrepancy” in our argumentation.
Round 2
Reviewer 1 Report
The revised manuscript is much easier to follow, and therefore is a valuable addition to the literature
Reviewer 2 Report
The authors answered all my comments.